# Selected Factors of Vascular Changes: The Potential Pathological Processes Underlying Primary Headaches in Children

**DOI:** 10.3390/children9111660

**Published:** 2022-10-30

**Authors:** Joanna Sordyl, Ilona Kopyta, Beata Sarecka-Hujar, Pawel Matusik, Tomasz Francuz, Ewa Malecka-Tendera

**Affiliations:** 1Department of Paediatrics, Faculty of Medical Sciences, Medical University of Silesia, 40-752 Katowice, Poland; 2Department of Paediatric Neurology, Faculty of Medical Sciences, Medical University of Silesia, 40-752 Katowice, Poland; 3Department of Basic Biomedical Science, Faculty of Pharmacy with the Division of Laboratory Medicine in Sosnowiec, Medical University of Silesia in Katowice, 41-200 Katowice, Poland; 4Department of Paediatrics, Paediatric Obesity and Metabolic Bone Diseases, Faculty of Medical Sciences, Medical University of Silesia, 43-100 Katowice, Poland; 5Department of Biochemistry, Faculty of Medical Sciences, Medical University of Silesia, 40-752 Katowice, Poland; 6Department of Paediatrics and Paediatric Endocrinology, Faculty of Medical Sciences, Medical University of Silesia, 40-752 Katowice, Poland

**Keywords:** children, atherosclerosis, dyslipidemia, headaches

## Abstract

Background: The prevalence, social consequences and complicated pathogenesis make headaches in children a significant clinical issue. Studies in adults suggest that primary headaches could be the first sign of atherosclerosis and platelet aggregation. Aim: To analyze the blood levels of selected biomarkers of vascular changes potentially associated with a higher risk of atherosclerosis in children with primary headaches. Methods: The medical family history, brain-derived neurotrophic factor (BDNF), soluble CD40 ligands (sCD40L), endothelial plasminogen activator inhibitor (PAI I), vascular endothelial growth factor (VEGF) and intima-media thickness (IMT) measurements were performed in the 83 children (52 with primary headaches, 31 controls). Selected factors were compared with basic laboratory parameters that are potentially related to atherosclerosis: C-reactive protein (CRP) and lipid concentration. Results: There were no significant differences in biomarkers of vascular changes in the study group and controls in general. In the study group, boys had a higher BDNF level than girls (*p* = 0.046). Normal-weight migraine patients had significantly higher PAI-I levels than controls (*p* = 0.034). A positive correlation between PAI-1 and triglycerides (TG) was observed. IMT did not differ between children with primary headaches and controls; however, IMT showed a positive correlation with BMI z-score and TG. Children with headaches had, more often, a positive family history of cardiovascular disease (*p* = 0.049). Conclusions: There were no clear clinical changes indicative of atherosclerosis in the study population. However, some trends are visible. Primary headaches are more often related to a family history of cardiovascular diseases. IMT is associated with TG levels and BMI z-score. The measured biomarkers of vascular changes show mutual relations.

## 1. Introduction

Headaches are one of the most common ailments that children and adolescents report to primary care physicians. It is estimated that the total frequency of headaches in children under the age of ten is 56%, while in adolescents it is 91%. Presumably, about 70% of school-age children experience headaches at least once a year [1,2,3,4]. This child’s ailment causes considerable anxiety, because it is often equated with symptoms of an ongoing neoplastic process or infections of the central nervous system. In fact, the mentioned, most serious causes, are only a small percentage of cases—primary headaches are predominant in developmental age groups [5,6].

Migraine is a disease of the central nervous system with several comorbidities, both central and systemic. Among the systemic ones, cerebro- and cardiovascular comorbidities have long been debated in recent years and clinical and subclinical atherosclerosis has been held responsible for this increased vascular risk in migraine patients [7]. In recent years, in addition to the commonly recognized risk factors for atherosclerosis, new markers of atherosclerotic lesions have been investigated more and more frequently. Brain-derived neurotrophic factor (BDNF), soluble CD40 ligand (sCD40L), endothelial plasminogen activator inhibitor (PAI I) and vascular endothelial growth factor (VEGF) are considered, potentially, the most useful for the clinical assessment of the severity of atherosclerotic lesions [8,9,10,11].

The aim of the work was to analyze the blood levels of selected biomarkers of vascular changes potentially associated with a higher risk of atherosclerosis in children with primary headaches compared to controls.

## 2. Materials and Methods

The study was prospective and it was approved by the Local Ethical Committee (KNW/0022/KB1/67/14, KNW/0022/KB1/67/I/14/16).

The population comprised 83 children (52 patients with primary headaches, 31 controls).

Children with primary headaches were recruited at the Department of Paediatric Neurology, the Medical University of Silesia in Katowice, Poland. The age at the recruitment was between 7 and 18 years of age. The analyzed group consisted of 32 girls (mean age 13.5 ± 3.6 years) and 20 boys (mean age 9.9 ± 2.6 years). The control group consisted of 31 children aged between 7 and 18 years old: 14 girls (mean age 10.0 ± 2.6 years) and 17 boys (mean age 10.8 ± 3.1 years). All patients from the control group were recruited from the Department of Paediatrics and Paediatric Endocrinology, the Medical University of Silesia in Katowice, Poland. Informed consent for participation in the study was obtained from the children’s parents/guardians and children older than 16 years.

The children included in the study group were assessed according to a uniform scheme, that is: a thorough assessment of the medical history, including family history, physical examination, neurological examination and neuroimaging by magnetic resonance imaging, or (in case of contraindications) using the computed tomography scan method. The inclusion criteria for the patients’ group were as follows: 7–18 years, a minimum of 2 episodes of headaches per week (migraine/tension-type headaches) for at least 6 months. The definition of primary headaches, both migraine and tension headaches, was adopted according to The International Classification of Headache Disorders 3rd edition (ICHD-3).

Patients were excluded in the case of: secondary headaches (fever, sinusitis, vision defect, glaucoma, brain tumors and other reasons for intracranial hypertension, such as inflammation of central nervous system diseases and/or hydrocephalus), acute respiratory tract or gastrointestinal infection, hormonal disorders (hypothyroidism, diabetes), acute/chronic disorders with hypertension/lipid disturbances- (renal and heart diseases, children born small for gestational age (SGA), familial hyperlipidemia), chronic medications use.

The children in the control group were diagnosed with constitutionally or family-related short stature and did not have headaches or other accompanying diseases. They were assessed according to a uniform scheme, that is: a thorough history of the disease was conducted, including family history, physical examination, neurological examination, and, in some justified cases, neuroimaging with magnetic resonance imaging or (in case of contraindications) computed tomography scan to exclude pituitary gland abnormalities. Body mass index (BMI) was calculated as weight (kg)/height (m^2^). To eliminate the effect of age and sex on the BMI body mass value, a normalized BMI z-score was calculated using a specific calculator (http://www.quesgen.com/BMIPedsCalc.php (accessed on 27 March 2013)). According to WHO, overweight was defined as BMI z-score > +1 SD (equivalent to BMI 25 kg/m^2^ at 19 years), obesity as BMI z-score > +2 SD (equivalent to BMI 30 kg/m^2^ at 19 years) and thinness as BMI z-score < −2 SD.

An Intima media thickness (IMT) measurement of the distal segment of the carotid arteries was performed by ultrasound using an Acuson Antares 10 Mhz linear head in B-mode. In order to ensure the best repeatability of the results, the tests were carried out by only one person—an experienced ultrasonography specialist. The thickness of the IMT complex was defined as the distance between two clearly shining distal wall lines. The first is the border between the lumen of the vessel and the inner membrane, the second sets the border between the middle membrane and the adventitia (Figure 1). IMT complex thickness measurements were made at several points (minimum 3) within the distal wall, about 1 cm proximal from the common carotid sinus (bifurcation for the internal and external carotid artery). The mean of all the measurements was taken as the result [12,13]. All activities were then repeated by conducting the study on the opposite side.

Blood samples were collected with approximately 12 h of fasting. After 2 h, the samples were centrifuged for further laboratory analysis.

Serum C-reactive protein (CRP) concentration was determined using the immunoturbidimetric method. Lipid parameters (total cholesterol (TC), high-density lipoprotein (HDL), low-density lipoprotein (LDL) and triglycerides (TG))were measured using enzymatic methods (commercial Pointe Scientific kits) immediately in fasting plasma. Brain-derived neurotrophic factor (BDNF), soluble CD40 ligand (sCD40L), endothelial plasminogen activator inhibitor (serpin E1/PAI I) and vascular endothelial growth factor (VEGF) concentrations were measured in serum by ELISA (R&D Systems) according to the manufacturer’s recommendations.

The STATISTICA 13 software (TIBCO Software Inc. (2017), StatSoft (Kraków, Poland), 30-110 Krakow. Statistica (data analysis software system), version 13. http://statistica.io (accessed on 1 October 2018)) was used to perform statistical analysis. For continuous variables, the mean values (M) and standard deviations (SD) were estimated. For categorical variables, absolute numbers (n) and relative numbers (%) were assessed. The normality of the distribution of quantitative data was evaluated by the Shapiro–Wilk W test. As these distributions differed from the normal distribution, to compare continuous variables, the U Mann–Whitney test was used. The stochastic independence χ^2^ test with Yates’s correction was used to compare categorical variables. The value of *p* ≤ 0.05 was considered to be statistically significant.

## 3. Results

The study group and the control group did not differ in terms of sex and the mean age of the patients. When analyzing the results obtained in the group of girls, it was observed that girls with headaches were statistically significantly older than asymptomatic girls (*p* = 0.025). A similar difference was also found within the same group of children with headaches—the mean age of girls was higher than that of boys (*p* = 0.005) (Table 1).

In most patients (83%), a family health history was irrelevant. Nevertheless, relatives of children with primary headaches had a more frequent occurrence of migraines or cardiovascular diseases, such as hypertension, ischemic heart disease, myocardial infarction and stroke with a borderline significance (*p* = 0.049). The presence of migraine in the family in all cases concerned the mothers and only in one case the mother and the siblings (Table 2).

The mean BMI z-score was statistically significantly higher in the study group compared to controls (0.0086 ± 1.140 vs. −1.364 ± 1.641; *p* = 0.00019). There were no statistically significant differences in the BMI z-score values between boys and girls, both within the study group and in the control group.

The prevalence of obesity and overweight in the groups was also analyzed. Seven of the headache patients were obese and one was overweight. Only one patient in the control group was overweight, and none were obese. However, these differences were not statistically significant.

In this study, the selected laboratory parameters were analyzed in relation to the occurrence of headaches, as well as sex and body weight.

The study group did not differ significantly from the controls in terms of the mean serum CRP concentration (0.717 ± 1.136 vs. 0.621 ± 0.659 (mg/L), *p* = 0.523).

There were no significant differences in the levels of biomarkers of vascular changes in the study group and controls in general. Nevertheless, some trends were noted (Table 3).

In the study group boys had a higher BDNF level than girls (mean 5473.068 ± 9900.325 vs. 11,771.93 ± 13,824.02 [pg/mL], *p* = 0.046). Similar differences due to sex were not observed in the control group.

Normal weight migraine patients had significantly higher PAI-I levels than controls (4228.29 ± 6525.067 vs. 9273.55 ± 7448.397 [ng/mL], *p* = 0.034).

Collective analyzes of all patients, both from study and control group, revealed that BDNF showed a negative correlation with sCD40L (r = −0.358, *p* = 0.00) and VEGF (r = −0.409, *p* = 0.00). VEGF showed a positive correlation with sCD40L (r = 0.656, *p* = 0.00), PAI-1- with the sCD40L (r = 0.361, *p* = 0.001) and VEGF levels (r = 0.426, *p* = 0.00).

Part of our results concerning lipid parameters in children with primary headaches was previously published in a separate study [14].

The only lipid parameter showing correlations with other markers was TG serum concentration. A positive correlation between PAI-1 and TG level was observed (r = 0.507, *p* < 0.05).

IMT did not differ between children with primary headaches and controls, however IMT showed positive correlation with BMI z-score (r = 0.386, *p* < 0.05) ant TG concentration (r = 0.391, *p* < 0.05).

In summary, the parameters that statistically significantly differed among children with headaches and controls were positive family medical history and BMI z-score. IMT showed a positive correlation with BMI z-score and TG concentration. Selected biomarkers of vascular changes exhibited mutual relations.

## 4. Discussion

Headaches in children are an important and complex issue with serious clinical and social consequences. Children suffering from chronic headaches report a lower quality of life than their healthy peers, maintained at a level comparable to chronic diseases such as arthritis or even cancer [15]. The approach to primary headache as a systemic disease leads to the search for new points of treatment and prevention.

Due to the numerous imperfections of the classical factors of atherosclerotic disorders, an intensive search for new markers useful in clinical practice is underway. In recent years, many publications considering the possible metabolic relationships of factors that may be involved in the processes of atherogenesis and platelet aggregation have been published. They include, inter alia, BDNF, sCD40L, serpin E1/PAI I and VEGF [8,9,10,11,16,17,18,19,20,21].

The analysis of the BDNF levels is performed due to its well-known role in pain sensitization and modulation in the central nervous system. BDNF secretion increases during a migraine attack [22,23]. BDNF values were also greater in the patients with tension-type headaches than in the control group. However, no differences between migraine patients with and without aura were found [23]. A possible point of BDNF’s influence on pain perception in the central nervous system is the interaction with the calcitonin gene-dependent peptide (CGRP) [24]. The processes of intracellular calcium homeostasis and signaling through the cell membrane calcium channel proteins are another important point of BDNF influence [25]. BDNF may downregulate the expression of superoxide dismutase and reduce the negative impact of free radicals [26].

Moreover, this protein may also be involved in pain pathophysiology through its participation in neurogenesis and synaptic maturation [27]. Presumably, BDNF plays an important role in the central nervous system by promoting neurite outgrowth and protecting neurons from apoptosis induced by serum deprivation [28].

Our study did not show any statistically significant differences in the BDNF concentration between the group of children with primary headaches and the control group. However, it was not checked whether the patients were in the active phase of the disease at the time of taking blood samples. Only among the study group, boys had higher BDNF concentrations than girls. Statistically significantly higher BDNF concentrations in boys, by promoting neuroplasticity and their potential in pain reduction, may be associated with a relatively lower incidence of primary headaches in the male gender.

Moreover, the effects of BDNF may not only be limited to the local effects on the central nervous system. Some authors speculate that this protein may be involved in systemic metabolism. Chaldakov et al. observed lower BDNF concentrations in adult patients with metabolic syndrome and atherosclerosis [29]. Hence, higher BDNF values may potentially show a beneficial metabotropic and vasoprotective effect [29]. However, these assumptions require further research, especially in the pediatric population.

Our data revealed that normal-weight migraine patients had significantly higher PAI-I levels than controls. This glycoprotein from the serpin family—serine protease inhibitors is produced by endothelial cells and could be secreted by other tissue types, such as platelets, hepatocytes and smooth muscle cells [30]. It is a recognized marker of thromboembolic disorders due to its significant role in the processes of fibrinolysis inhibition [31]. There are a number of factors regulating PAI-1 secretion, including those that directly affect PAI-1 gene transcription (IL-1 and IL-6, TNF-α, TGFβ (transforming growth factor β), EGF, fibrin esters, insulin and lipoprotein of VLDL fraction) [32,33,34]. Considering the fact that many of these substances are also involved in the pathophysiology of primary headaches, it seems reasonable to investigate the possible role of PAI-1 in the development of these ailments. In this study, no statistically significant differences in the levels of this protein between children from the test and control groups were found. However, we showed that high PAI-1 concentrations correlated positively with both sCD40L and VEGF levels. The available literature data are sparse and based mainly on the analysis of adult patients. It has been observed that PAI-1 levels may be reduced in adult patients suffering from migraines [35]. Moreover, the frequency of homozygotes and heterozygotes polymorphism for PAI-1 as well as other prothrombotic factors were demonstrated in the group of adult women with migraine [36]. PAI-1 may also be involved in the pathophysiology of migraine through the effect of ischemia. Hypoxia is one of the postulated triggers for migraine and aura as it is able to induce migraine attacks and aura independently of any pharmacological agent [37]. On the other hand arterial oxygen desaturation leads to decreased expression of plasminogen activators and increased secretion of PAI-1 which results in blood hypercoagulability and microembolism [38,39,40]. A large and representative Icelandic sample showed a correlation between migraines with an increased risk of ischemic stroke in adult patients in comparison to the healthy population [41,42]. Studies performed on children did not provide strong conclusions.

Another factor potentially involved in the primary headache pathophysiology is VEGF. It is involved in angiogenesis and tissue remodeling [43]. The most important factors causing the increased secretion of VEGF are hypoxia, growth factors and various types of inflammatory cytokines. VEGF plays a significant role in the processes of angiogenesis, not only in physiological conditions but also in a number of diseases such as cancer, rheumatoid arthritis, psoriasis or diabetic retinopathy [44,45]. The loss of a single VEGF allele may results in serious lethal cardiovascular defects [46]. Therefore, it is presumed that the VEGF gene polymorphism may play a role in the pathomechanism of vascular diseases, such as migraines. According to Goncalves et al., VEGF haplotypes are associated with primary headache susceptibility in migraineur women compared to healthy controls [47].

Presumably, the role of VEGF in organism homeostasis seems to be wider than previously reported. Recently, several studies indicating the influence of this protein on the body’s energy balance and the function of the adipose tissue have been published. Animal studies have shown that an increased VEGF expression may protect against insulin resistance and excessive energy intake obesity. Some authors reported that higher VEGF concentrations correlated with higher BMI in adult patients [48,49]. However, in this study, no correlation between BMI and the concentration of VEGF was found.

In our study, a high BDNF value was associated with lower sCD40L concentrations. The opposite tendency was shown in the relationship between VEGF and sCD40L—the higher the VEGF values, the higher the sCD40L concentrations were. sCD40L, a ligand of the glycoprotein IIb-IIIa receptor, is well-known as a factor involved in inflammation, immunity and platelet activation. Its role is to stabilize the thrombus and platelet activation [50,51,52]. The results of some recent studies indicated that it may also be an independent marker of developing atherosclerotic lesions. Schonbeck et al. [51]. found that high levels of sCD40L were associated with a higher risk of heart attack, stroke or cardiovascular death in apparently healthy middle-aged women—regardless of smoking. Moreover, it seems that the concentrations of this marker measured in peripheral blood are very similar to the values recorded in samples taken directly from the coronary arteries in patients with acute coronary syndrome [53]. These clinical properties of sCD40L potentially make it a convenient and reliable marker of subclinical atherosclerotic lesions linking some diseases, such as primary headaches, with vascular disorders. Guldiken et al. reported higher sCD40L in the interictal period in adult migraine patients compared to controls [50]. The results obtained in this study did not confirm the existence of differences in sCD40L concentration between children with primary headaches and controls. However, the observed relationships between this factor and BDNF and VEGF seem to confirm the assumptions that high concentrations of VEGF and sCD40L may be associated with potentially unfavorable metabolic parameters, while BDNF shows a rather protective effect.

Among the analyzed classic laboratory parameters and new biomarkers of vascular disturbances, only PAI-1 and TG showed statistically significant interrelationships. Dyslipidemia is a common, well-known risk factor for cerebrovascular and cardiovascular diseases [54,55]. Associations between lipid disturbances and the occurrence of headaches have already been shown in adult patients. The results obtained in studies in children are limited and are mainly based on the relationship between abnormal total cholesterol (TC) and low-density lipoprotein (LDL) levels [56,57,58]. However, it should be emphasized that in the pediatric population the greatest emphasis is placed on the concentrations of TG and HDL cholesterol as the cardiometabolic risk factors. TG levels above the 75th percentile and HDL levels below the 25th percentile for sex and age may be associated with an unfavorable prognosis [59,60]. It was also demonstrated that the level of serum triglycerides is a risk factor for ischemic stroke in children [61,62,63]. Nevertheless, there are also a number of studies that do not support the association of headaches in children with lipid disturbances [64].

## 5. Limitations of the Study

We made every effort to ensure that both the study and control groups were selected from patients with a uniform profile, but we are aware of the weak sides of the work. The most important is that although the diagnoses were made on the basis of the ICHD-3 criteria, the data of the patients with different types of primary headaches were interpreted together. This was an important factor that potentially affected the obtained results, especially in the context of the relationships between the biomarkers of vascular changes. To fully interpret the data, it is necessary to expand the study group so that analyzes can be performed in the subgroups. Moreover, if more patients with spontaneous headaches were collected, it would be possible to distinguish age subgroups, for example early school children and adolescents. The authors plan to expand the study group and carry out the above-mentioned analysis, taking into account also the potential influence of for example the child’s age or puberty degree of the endothelial damage. A similar remark concerns the influence of sex on the parameters tested, as well as body weight.

## Figures and Tables

**Figure 1 children-09-01660-f001:**
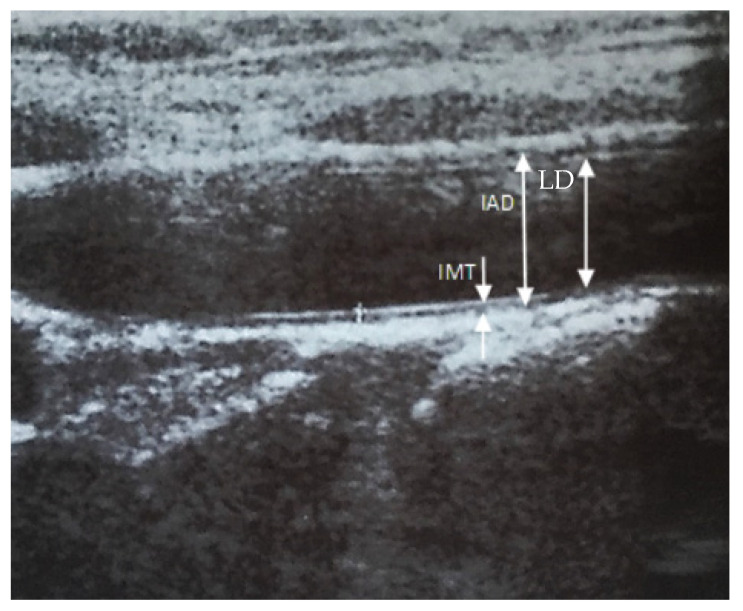
Diagram of the common carotid artery in ultrasound examination. IMT—intima media thickness, IAD—interadventitial diameter; LD—lumen diameter.

**Table 1 children-09-01660-t001:** Characteristics of the study group (SG) and control group (CG).

	SG *n* = 52	CG *n* = 31	
Girls (%) Boys (%)	32 (61.5 %) 20 (38.5%)	14 (45.2%) 17 (54.8)	*p* = 0.146
Age—total (years), mean ± SD Age—girls (years), mean ± SD Age—boys (years), mean ± SD	12.078 ± 3.720 13.519 ± 3.622 9.869 ± 2.629	10.886 ± 2.848 11.010 ± 2.580 10.784 ± 3.126	*p* = 0.177 *p*= 0.025 *p* = 0.407

**Table 2 children-09-01660-t002:** Characteristics of a family health history in the study group (SG) and control group (CG).

Family Health History	SG	CG	Total	*p*
Migraine (%)	4 (4.8 %)	0	4	0.049
Cardiovascular diseases (%)	4 (4.8 %)	0	4	
Irrelevant (%)	38 (45.78 %)	31 (37.4%)	69	
No data (%)	6 (7.22 %)	0	6	
Total (%)	52 (62.6 %)	31 (37.4%)	83	

**Table 3 children-09-01660-t003:** Mean values of selected biomarkers of vascular changes in the study group (SG) and the control group (CG).

	SG	CG	
BDNF [pg/mL], mean; median	8211.703; 56.50	17,785.85; 20,016.46	*p* = 0.071
VEGF [pg/mL], mean; median	127.632; 56.67	85.75; 48.63	*p* = 0.739
sCD40L [pg/mL], mean; median	1319.06; 503.72	1030.91; 560.89	*p* = 0.308
PAI-1 [pg/mL], mean; median	5059.29; 224.62	9273.55; 10,557.77	*p* = 0.659

## Data Availability

Not applicable.

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
