# Peer review of "Selected Factors of Vascular Changes: The Potential Pathological Processes Underlying Primary Headaches in Children"

_children, 2022, doi:10.3390/children9111660_

Round 1

Reviewer 1 Report

First of all, I suggest replacing 'idiopathic headache' with 'primary headache' throughout the article. This is more in line with what is written in the International Headache Classification.

On line 49, the authors should remodel their statements as follows. Migraine is a disease of the central nervous system with several comorbidities, both central and systemic. Among the systemic ones, cerebro- and cardiovascular comorbidities have long been debated in recent years and clinical and subclinical atherosclerosis has been held responsible for this increased vascular risk in migraine patients.

At line 77 and line 93, the authors write that children (patients and controls) underwent MRI or CT scans. Why? In the diagnostic process of primary headache, I am not aware of any provision to subject everyone to neuroimaging, the same for healthy subjects.

The article lacks the diagnosis of primary headache. Once the diagnosis of primary headache (migraine, tension-type headache, etc.) has been made, patients should be analysed according to the individual diagnosis, not mixed all together. Idiopathic headache can mean more than 40 different headache types with different pathophysiologies and possible cerebro-cardio-vascular involvement.

On line 124, I am not aware that Statsoft owns Statistica v13.0.

In the results: Figure 2 does not seem to be necessary.

In light of the negative results of this manuscript, the discussion should be greatly reduced: it is too tangled, long and speculative and should therefore be reduced to a maximum of 2 pages. Authors should only discuss the results in relation to the "AIM" of their work, specified at the end of the Introduction (lines 58-60), and nothing else. Furthermore, the beginning of the discussion should be devoted to the description of the (negative) results, then a point-by-point discussion of the (negative) results should take place in the light of the previous articles.

Author Response

Author's Response to Decision Letter for

“Selected factors of vascular changes: The potential pathological processes underlying idiopathic headaches in children”

October, 17th, 2022

Dear Editor-in-Chief
Children,
Thank You for considering the publication of our paper, “Selected factors of vascular changes: The potential pathological processes underlying idiopathic headaches in children” in Children. We were very grateful for the quick response. Authors are confident that all corrections suggested by Reviewers enriched our manuscript and made it more useful in clinical practice. Please see below for our responses to Referees‘  comments. You will also find all specific edits made in red coloured text and track changes noting removal of text, throughout our resubmitted manuscript.

We appreciate Your reconsideration of our revised manuscript and hope that it is now suitable for publication. 

Kind Regards, 

Sincerely, 
Ilona Kopyta
Medical University of Silesia in Katowice, School of Medicine in Katowice, Faculty of Medical Sciences, Department of Paediatric Neurology

RESPONSE TO REVIEWER 

Comment: “First of all, I suggest replacing 'idiopathic headache' with 'primary headache' throughout the article. This is more in line with what is written in the International Headache Classification.”

Answer: Thank You for suggesting changing the term "idiopathic headache" to be more in line with the current literature. All these phrases have been replaced by "primary headache" in the text.

Comment:On line 49, the authors should remodel their statements as follows. Migraine is a disease of the central nervous system with several comorbidities, both central and systemic. Among the systemic ones, cerebro- and cardiovascular comorbidities have long been debated in recent years and clinical and subclinical atherosclerosis has been held responsible for this increased vascular risk in migraine patients.”

Answer: We appreciate your help in re-editing this text to be clearer and more precise. It has been revised in the manuscript according to your suggestion.

Comment: “At line 77 and line 93, the authors write that children (patients and controls) underwent MRI or CT scans. Why? In the diagnostic process of primary headache, I am not aware of any provision to subject everyone to neuroimaging, the same for healthy subjects.”

Answer: The authors fully agree that there is no obligation to perform neuroimaging in all subjects in the diagnostic process of primary headache. In our Center, a large number of children with primary headaches are conducted on an outpatient basis without unnecessary and excessive diagnostic tests. Nevertheless, the study group was recruited among the patients of the Neurology Department, where children with headaches were referred in order to deepen the diagnosis - mainly to exclude the causes of secondary headaches. It seems to us that the inclusion of patients with normal neuroimaging results allows for a better selection of patients in the study group and avoiding the potential bias of the results.

As for the patients in the study group, neuroimaging was performed only in some justified cases in order to exclude pituitary gland abnormalities. This fact was specified in the manuscript.

Comment: The article lacks the diagnosis of primary headache. Once the diagnosis of primary headache (migraine, tension-type headache, etc.) has been made, patients should be analysed according to the individual diagnosis, not mixed all together. Idiopathic headache can mean more than 40 different headache types with different pathophysiologies and possible cerebro-cardio-vascular involvement.

Answer: The authors are aware of this limitation of the study. The study originally assumed an analysis according to the individual diagnosis. Due to the small size of the subgroups, it was decided to analyze all headache patients together. We plan to extend the study to include the assessment of the analyzed parameters in the subgroups depending on the diagnosis.

Comment: On line 124, I am not aware that Statsoft owns Statistica v13.0.

Answer: The program specification has been checked and supplemented in the text.

Comment: In the results: Figure 2 does not seem to be necessary.

Comment: In light of the negative results of this manuscript, the discussion should be greatly reduced: it is too tangled, long and speculative and should therefore be reduced to a maximum of 2 pages. Authors should only discuss the results in relation to the "AIM" of their work, specified at the end of the Introduction (lines 58-60), and nothing else. Furthermore, the beginning of the discussion should be devoted to the description of the (negative) results, then a point-by-point discussion of the (negative) results should take place in the light of the previous articles.

Reviewer 2 Report

Please add if the data set was checked for normal distribution and which test was used and if the data was normal or not. This is to ensure parametric and non-parametric choices for comparison.

Please add about the power of the study. Did the author consider the sample size calculation before the trial and checked the power again after finalizing the study? This is to ensure a sufficient sample size and that the study is not underpowered.

The discussion reads very long. If possible, add a subheading for biomarkers to break the long text into pieces easier to follow and understand. A summary table can also be helpful.

Author Response

Author's Response to Decision Letter for

“Selected factors of vascular changes: The potential pathological processes underlying idiopathic headaches in children”

October, 17th, 2022

Dear Editor-in-Chief
Children,
Thank You for considering the publication of our paper, “Selected factors of vascular changes: The potential pathological processes underlying idiopathic headaches in children” in Children. We were very grateful for the quick response. Authors are confident that all corrections suggested by Reviewers enriched our manuscript and made it more useful in clinical practice. Please see below for our responses to Referees‘  comments. You will also find all specific edits made in red coloured text and track changes noting removal of text, throughout our resubmitted manuscript.

We appreciate Your reconsideration of our revised manuscript and hope that it is now suitable for publication. 

Kind Regards, 

Sincerely, 
Ilona Kopyta
Medical University of Silesia in Katowice, School of Medicine in Katowice, Faculty of Medical Sciences, Department of Paediatric Neurology

RESPONSE TO REVIEWER 

Comment: Please add if the data set was checked for normal distribution and which test was used and if the data was normal or not. This is to ensure parametric and non-parametric choices for comparison.

Answer: The normality of distribution of quantitative data was evaluated by the Shapiro-Wilk W test. The analyzed distributions differed from the normal distribution in all cases.
The missing information has been completed in the manuscript.

Comment: Please add about the power of the study. Did the author consider the sample size calculation before the trial and checked the power again after finalizing the study? This is to ensure a sufficient sample size and that the study is not underpowered.

 Answer: The Authors did not perform the sample size calculation. The study was prospective and all the patients hospitalized in the Neurology Department during the study period who fulfilled inclusion criteria as well as who gave written informed consent were included in the study group. We are aware that this is a limitation of the study. Calculating the power of the study will be considered in planning extension of the study.

Comment: The discussion reads very long. If possible, add a subheading for biomarkers to break the long text into pieces easier to follow and understand. A summary table can also be helpful.

Answer: Thank you for this suggestion. The authors have made every effort to remove unnecessary, confusing fragments of the text and make the discussion clearer and focused on the purpose of the work. The discussion has been shortened to about 2 pages and now contains only information directly related to the aim of the work.

Round 2

Reviewer 1 Report

I am happy that the authors recognise the big problem of including patients with different diagnoses. At the same time, they absolutely must include this limitation (lack of proper diagnosis accordig to the International Classificazion of Headache Disorders) in the final section of the manuscript, making it clear that it is precisely this major confounding factor that could be behind the study's substantially negative results regarding the biomarkers of vascular change and IMT values.

Author Response

Author's Response to Decision Letter for

“Selected factors of vascular changes: The potential pathological processes underlying idiopathic headaches in children”

October, 20th, 2022

Dear Editor-in-Chief
Children,

Thank You that our paper  “Selected factors of vascular changes: The potential pathological processes underlying primary headaches in children” has been re-reviewed by the experts in the field.

We were very grateful for the quick response and valuable comments on the content of the work. Authors made every effort to make all the corrections suggested by the Reviewers.

Please see below for our responses to Referees‘  comments. You will also find all specific edits made in red coloured text and track changes noting removal of text, throughout our resubmitted manuscript. 

We appreciate Your reconsideration of our revised manuscript and hope that it is now suitable for publication. 

Kind Regards, 

Sincerely, 
Ilona Kopyta
Medical University of Silesia in Katowice, School of Medicine in Katowice, Faculty of Medical Sciences, Department of Paediatric Neurology

RESPONSE TO REVIEWER 

Comment: “I am happy that the authors recognise the big problem of including patients with different diagnoses. At the same time, they absolutely must include this limitation (lack of proper diagnosis according to the International Classification of Headache Disorders) in the final section of the manuscript, making it clear that it is precisely this major confounding factor that could be behind the study's substantially negative results regarding the biomarkers of vascular change and IMT values.”

Answer: The authors would like to thank You once again for pointing out the need for a clearer indication of the limitations of the study in the text. We made every effort to supplement the final section of the manuscript with the information that, although the diagnoses were made on the basis of the ICHD-3 criteria, the data of the patients with different diagnoses were interpreted together. We tried to emphasize that this was a factor that potentially affected the obtained results and it is necessary to expand the study group so that analyzes can be performed in the subgroups to fully interpret the data.
